# How Does Trait Mindfulness Weaken the Effects of Risk Factors for Adolescent Smartphone Addiction? A Moderated Mediation Model

**DOI:** 10.3390/bs13070540

**Published:** 2023-06-28

**Authors:** Dengfeng Li, Yang Xu, Shangqing Cao

**Affiliations:** 1School of Journalism and Communication, South China University of Technology, Guangzhou 510006, China; dengfeng@mail.scut.edu.cn (D.L.); 202220152682@mail.scut.edu.cn (Y.X.); 2School of International Education, South China University of Technology, Guangzhou 510006, China

**Keywords:** trait mindfulness, smartphone addiction, social anxiety, left-behind experience

## Abstract

As a psychological resource of individuals, trait mindfulness is valuable in facilitating individuals to maintain attention intensity, increase efficiency, and alleviate stress and depression. It can also buffer against the risk factors of addictive behaviors. However, applied research combining trait mindfulness and smartphone addiction with the use of psychological resources is relatively scarce and needs further examination. We constructed a moderated mediation model based on compensatory Internet use and conservation of resources theory (OCR) to examine the effects of social anxiety on adolescent smartphone addiction and to describe how trait mindfulness “works” and “in what contexts it works better”. We analyzed 1570 adolescent subjects through a multistage stratified sampling method. Our findings revealed that social anxiety positively predicted smartphone addiction, while trait mindfulness was negatively associated with it. Furthermore, trait mindfulness mitigated smartphone addiction by reducing social anxiety, suggesting a mediating effect of social anxiety on this relationship. Meanwhile, the mediating effect was more pronounced among adolescents with left-behind experience; we found that left-behind experience partially moderated the relationship between social anxiety and smartphone addiction. Adolescents with left-behind experience had more significant compensatory media use with a higher risk of smartphone addiction. This study highlights the potential protective role of trait mindfulness in the development and maintenance of adolescent smartphone addiction. It provides empirical support for applying resource conservation theory and stress buffering theory in this context.

## 1. Introduction

In the past five years, the number of Internet users and the Internet penetration rate in China have risen sharply. As of December 2022, the proportion of Internet users using mobile phones as Internet terminals was as high as 99.6%, with an average weekly Internet access time of 29.4 h. It is worth noting that the number of Internet users among minors has reached 191 million, and the Internet penetration rate has reached 96.8%. The Internet penetration rate among children is much higher than the overall level, and the trend of exposure to the Internet at a younger age is more significant [1], which has also given rise to a more serious phenomenon of smartphone addiction. According to a report released by the China Youth Network Association 2021, the detection rate for smartphone addiction is the highest among adolescents aged 13–18 in China, reaching 17.1%. Unlike substance addiction, smartphone addiction is purely a behavioral or psychological addiction, entailing excessive smartphone use behavior and continuous dependence on and craving for smartphones, including for behavioral and emotional support [2]. In addition, smartphone addiction is accompanied by a lack of ability to regulate and control one’s smartphone use, which hurts daily life [3]. This phenomenon in the adolescent population is particularly pronounced.

The existing literature on smartphone addiction among adolescents has predominantly focused on investigating the “formation mechanism” and “solution countermeasures”. Empirical studies have extensively explored the antecedents of smartphone addiction and revealed that personality factors, such as self-concept clarity, fear of missing out, and alexithymia, significantly predict smartphone addiction [4,5]. Moreover, environmental factors, including parental phubbing, childhood abuse, and other risks, have been identified as potential contributors to adolescent smartphone addiction [6,7,8]. In terms of countermeasures, attentional and cognitive training, motor interventions, parental supervision, and modeling effects have received considerable attention in the literature [9,10,11]. Empirical evidence has suggested that smartphone addiction behaviors are closely linked to individual emotions and psychological cognition [12,13]. Therefore, it is crucial to identify the factors that influence and mitigate the risk of smartphone addiction and to elucidate the underlying mechanisms. By comprehensively examining these factors, we can develop evidence-based interventions to prevent and treat smartphone addiction among adolescents.

Prior research has identified social anxiety as a potential risk factor for smartphone addiction and has proposed corresponding intervention strategies [14], thereby providing a valuable empirical reference for the present study. However, the underlying mechanisms remain unclear. Trait mindfulness has been identified as a potential individual psychological resource that may counteract social anxiety, but the specific mechanisms and contexts of its efficacy require further investigation. To address these gaps, we examined a sample of adolescents aged 13–18 years in the western region, including left-behind and non-left-behind children. Drawing on compensatory Internet use and conservation of resources theory, we constructed a mediation-moderation model that integrated trait mindfulness, social anxiety, and other relevant variables. Specifically, we aimed to address two key issues: first, whether trait mindfulness can serve as a buffer against the negative effects of smartphone addiction and the underlying mechanisms; second, the role of left-behind experience in the relationship between trait mindfulness and adolescent smartphone addiction. Our study contributes to a better understanding of the mechanisms underlying the link between social anxiety and smartphone addiction, particularly for those with left-behind experience. Moreover, our findings provide targeted theoretical and empirical references for identifying the risk factors and intervention strategies for internalization problems associated with adolescent smartphone addiction.

## 2. Literature Review

### 2.1. Trait Mindfulness and Smartphone Addiction

Mindfulness originated from Eastern Buddhist meditation training and refers to the intensity of attention. It was refined, processed, and integrated by the Western psychological community in the late 1970s into a mental construct that blends Buddhist meditation with psychological thought. Mindfulness is a multidimensional concept that can refer to mindfulness training or interventions and to trait or state mindfulness [15]. Research indicates that state mindfulness is a professional psychological quality acquired through mindfulness training, meaning that mindfulness is interventional and modifiable [16]. In contrast, trait mindfulness is the ability of an individual to remain aware of and focused on the experience of the present moment. It can be viewed as a stable state of mind that requires the individual to consciously focus on a fixed object. In short, it can be referred to as a mental resource for individuals. To explore the effects of sustained rather than transient behavior, we focus on trait mindfulness, emphasizing the average level of mindfulness in an individual’s daily life.

According to the resource conservation theory, individuals tend to cope with anxiety and stress by acquiring, investing in, or safeguarding resources they value. The term “resources” here encompasses any abilities, objects, personality traits, energy, or conditions that people find valuable [17]. People tend to conserve resources and use them to acquire other resources, and the loss or gain of resources depends on the possession of other resources [18]. As a psychological resource for individuals, the gaining of trait mindfulness has received extensive attention. Research has indicated that trait mindfulness can help individuals maintain good mental health, improve subjective well-being [19], enhance attentional intensity and mental clarity [20], reduce stress [21] and depressive mood [22], and serve as an effective treatment option for anxiety disorders [23], indicating its high practical value. Nonetheless, applied research that combines trait mindfulness with smartphone addiction and explores the psychological resources of adolescents is relatively scarce.

Previous studies suggest that trait mindfulness can promote individual psychosocial adjustment and be used to treat various psychiatric disorders, including behavioral addiction and dependence [24]. Mindfulness levels are negatively correlated with smartphone addiction before sleep [25]. Moreover, mindfulness levels have a dampening effect on risk factors for problematic smartphone use, and when mindfulness is enhanced, the impact of fear and boredom tendencies on smartphone addiction is reduced. A case study showed that mindfulness training improved smartphone addiction in students [26], and mindfulness meditation significantly alleviated uncontrolled reactions, withdrawal, and inefficiency related to college students’ smartphone addiction [23]. This is because mindfulness therapy achieves its efficacy through psychological mechanisms, such as emotion regulation, cognitive reappraisal, and self-compassion. This also supports resource conservation theory’s view that trait mindfulness enables people to focus their cognitive attention on the present moment [27], manage complex emotions more effectively, cope with risks, and achieve greater well-being in daily life. Additionally, mindfulness training enables individuals to maintain an open and accepting attitude, develop a greater ability to “let go” of negative experiences [28], and healthily and beneficially focus on experience, effectively inhibiting addictive behaviors.

Drawing on resource conservation theory, we suggest that trait mindfulness can help individuals cope with risky behaviors and accordingly propose the following research hypothesis. 

**Hypothesis** **1.**
*Individuals with higher trait mindfulness have richer psychological resources to cope with risky behaviors; i.e., trait mindfulness negatively affects smartphone addiction.*


### 2.2. The Mediating Role of Social Anxiety

Social anxiety was initially introduced by British psychiatrist Mark, who distinguished individuals who fear social situations due to phobic disorders and called this emotional symptom “social anxiety”. This group tends to show more struggles in interpersonal interactions, with intense apprehension, stressful and emotional reactions, and avoidance behaviors in interpersonal situations [29]; difficulties in establishing and maintaining good interpersonal relationships; and even multiple negative effects, such as poor sleep, academic burnout, and depressive symptoms. In addition, social anxiety tends to produce more sensitive stress responses, such as greater physiological arousal and more redundant thoughts and other non-random responses. Thus, social anxiety tends to decrease the organizational resources available to cope with stress, such as impairing executive functioning and reducing self-regulation. Thus, adolescents with social anxiety tend to report more avoidance behaviors and less use of productive coping styles.

Compensatory Internet use theory postulates that individuals who are unable to satisfy their psychological needs in real-life settings may seek compensatory satisfaction through online platforms. The rationale behind this theory is that individuals who face difficulties with offline face-to-face communication may try to compensate for their shortcomings using online means [30]. Compared to offline communication, online communication offers a degree of anonymity and reduced emphasis on physical appearance, which may result in less stress and greater control over the communication process. Smartphones, in particular, provide a more comfortable social environment for adolescents who struggle to express themselves in real life [31]. Consequently, individuals who have trouble communicating in person or have difficulty socializing offline, such as those with negative self-esteem [32], high social anxiety [33], and loneliness [34], are more inclined to use mediated channels for communication. However, it is important to note that using digital media with compensatory motives can sometimes have adverse outcomes and negative consequences. Individuals whose social needs are fulfilled through smartphone use may develop a reliance on these devices, leading to addictive tendencies [35,36,37]. Social media use, in particular, has been linked to depression and other psychological issues [38,39]. Given that social anxiety is recognized as a significant risk factor for developing smartphone addiction, interventions for smartphone addiction should consider social anxiety as a relevant factor that needs attention.

Mindfulness is a pathway to self-regulating attention while being accepting of and open to one’s experiences in the present moment [15]. It positively contributes to individual emotion regulation and behavioral interventions. Although mindfulness-based interventions (MBIs) focus on operational changes in attention and awareness, mindfulness has also been conceptualized as a dispositional trait. Individuals with high levels of trait mindfulness tend to be nonreactive and nonjudgmental about their internal experiences; observe their cognitions and emotions; and demonstrate intentional awareness and focused action in daily activities [40]. Previous research has shown that trait mindfulness is associated with a tendency to relieve stress and reduce addictive behaviors. Trait mindfulness levels promote positive emotions and life satisfaction and reduce negative emotions, such as anxiety, depression, and aggression [41].

In addition, it has been suggested that trait mindfulness can act as a buffer against the deleterious effects of risk factors associated with addictive behaviors, including perceived stress [41], anxiety [42], and craving [43]. Similarly, trait mindfulness may be a protective factor against smartphone addiction, potentially reducing the impact of its risk factors. Specifically, a study revealed that the negative relationship between trait mindfulness and compulsive social media use was mediated by social anxiety [44]. In other words, trait mindfulness also has lowering effects on stress derived from such compulsive behavior, mediated by the former variables. This research indicated that trait mindfulness may serve as a safeguard against the adverse effects of negative emotional states. Taken together, this demonstrates that the effects of cognitive and affective risk factors on smartphone addiction depend on one’s mindfulness [45].

Therefore, with trait mindfulness serving as the starting point, we propose the following research hypothesis.

**Hypothesis** **2.**
*Social anxiety mediates the relationship between trait mindfulness and smartphone addiction. Specifically, trait mindfulness may influence smartphone addiction by reducing individuals’ social anxiety levels.*


### 2.3. The Moderating Effect of the Left-Behind Experience

Survey data show that there are more than 12 million left-behind children in compulsory education in China, and 96% of left-behind children are supervised by grandparents and 4% by other relatives and friends. Empirical studies have shown that left-behind children have poor mental health status and present a complex state of emotional entanglement between closure and longing. Inadequate positive emotional experiences, blocked negative emotional expression, and poor emergency emotional communication are the main problems in their families’ emotional lives [46], and emotional issues are usually associated with mental health and risky behaviors. While most existing studies have focused on the current status of left-behind children, there is a lack of exploration of the potential impact of left-behind experience in childhood on subsequent psychological development and protective factors. Based on this, we focused on the impact of left-behind experience on addictive behaviors in early adulthood with a developmental perspective.

Adolescence is a critical developmental period that plays a pivotal role in shaping positive psychological qualities and personality traits among adolescents with left-behind experience [47]. However, adolescents with left-behind experience are prone to emotional maladjustment and various transgressive behaviors due to prolonged parent–child separation, lack of emotional companionship, and limited care and supervision from temporary guardians, as well as inadequate support from important developmental environments, such as schools and communities [48]. Empirical evidence suggests that adolescents with left-behind experience exhibit lower levels of psychological resilience and a higher prevalence of mental health issues, including anxiety, depression, loneliness, and somatization, in comparison to their non-left-behind counterparts [49], thus indicating a plausible association between left-behind experience and mental health concerns. Such an association has also been found to be conducive to other psychological and behavioral issues, such as Internet addiction. The early upbringing of left-behind children, characterized by indifferent or neglectful parenting, diminishes their psychosocial resources, such as self-esteem and social support, thereby impairing their psychological well-being. In contrast, non-left-behind children tend to live in a better family environment with intact social support. They tend to use optimistic attitudes to meet challenges, enhancing the protective effect against psychological and behavioral problems.

Although previous studies have affirmed the mediating role of social anxiety between individual psychological traits and addictive behaviors [50], there are some inconsistent results, such as some researchers reporting that the two variables are only moderately positively correlated [51,52], while others have found that psychological traits and smartphone addiction are highly positively correlated [53]. This suggests that the mediating mechanisms by which social anxiety is mediated are moderated by several contextual factors. In addition, related studies have also shown that left-behind experience is an important moderating variable in the relationship between psychological resources and individual adjustment problems. Thus, we propose the following research hypothesis.

**Hypothesis** **3.**
*Left-behind experience may also moderate the mediation process of trait mindfulness in relation to smartphone addiction.*


Drawing on the above information, this study proposes a research pathway to investigate adolescent smartphone addiction starting from trait mindfulness. Specifically, we incorporate left-behind experience to construct a mediated model (Figure 1) that aims to elucidate the mechanism by which trait mindfulness influences adolescent smartphone addiction and to provide a more comprehensive understanding of the formation mechanisms underlying adolescent smartphone addiction.

## 3. Materials and Methods

### 3.1. Questionnaire Design

In this study, we used a self-assessment questionnaire, which was designed in two parts. The first part collected basic demographic data from the participants, such as gender, age, and grade, which were treated as control variables in the analyses. Furthermore, based on the definition of left-behind children, we identified participants who had experienced left-behind situations and assigned a value of 1 to them, while those who had not were assigned a score of 0. This variable was used to test for moderating effects. The second part of the questionnaire used established scales to measure the study variables. We sought the help of two English graduate students to translate the English scales into the study context, and professional psychometricians provided feedback to improve the scales.

### 3.2. Measurement of Variables

#### 3.2.1. Trait Mindfulness

The measurement of trait mindfulness in this study utilized the Mindfulness Attentional Awareness Scale (MAAS), which was originally developed by Brown and Ryan and later revised by Deng et al. [54]. This 15-item scale uses a 6-point Likert-type response scale ranging from 1 (rarely) to 6 (always) to assess an individual’s level of mindfulness. Reverse scoring was applied to all items, and the total score was calculated by summing the scores of all items. Higher scores indicate a higher level of trait mindfulness. The internal consistency of the scale was evaluated using Cronbach’s alpha coefficient, which was found to be 0.882 in this study.

#### 3.2.2. Smartphone Addiction

The Mobile Phone Addiction Tendency Scale for College Students (MPATS) developed by Xiong J. et al. [55] was used to assess smartphone addiction in this study. The scale includes 16 items and measures 4 dimensions comprising withdrawal symptoms, emergent behaviors, social soothing, and mood changes. Responses are given on a 5-point Likert scale ranging from 1 to 5, with higher scores indicating more serious smartphone addiction. The scale has a high internal consistency, with Cronbach’s alpha coefficients of 0.935 for the overall scale and 0.830, 0.760, 0.808, and 0.745 for the four dimensions, respectively.

#### 3.2.3. Social Anxiety

The Social Anxiety Scale was administered using a version developed by Scheier and Carver [56], which was more appropriate to the study context. The scale measures subjective anxiety, verbal expressions, and behaviors and assesses the experience of subjective anxiety in interpersonal interactions. One of the questions was reverse-scored. In this study, the Cronbach’s alpha coefficient of the social anxiety scale was 0.852, indicating the high internal consistency of the scale.

### 3.3. Data Collection and Description

The present investigation was conducted in four cities situated in the western region of China, which were deemed to be representative labor-export cities characterized by significant urban–rural differences and a conspicuous prevalence of the left-behind children phenomenon. Prior to the official survey, a pre-survey reliability test was administered to a cohort of 50 adolescents from city H, with the scale’s Cronbach’s alpha coefficient amounting to 0.7. Based on feedback, the scale entries and the wording of the questionnaire were revised. The formal survey, which adhered to the principles of confidentiality and informed consent, was carried out in February 2023 across eight middle schools located in the four cities of Guangxi Zhuang Autonomous Region. A multistage stratified sampling method was utilized, with district and grade level being utilized as the basis of stratification and the whole class being sampled to ensure uniform distribution of the questionnaire. A group reading of the introduction was conducted, followed by the on-site collection of the questionnaire, which took approximately 20 min to complete.

A total of 1754 questionnaires were collected, and following a thorough examination of missing data, incomplete responses, and logical inconsistencies, 1570 valid questionnaires were retained, corresponding to an effective rate of 89.51%. The respondents were mostly female (*N* = 921, 58.7%). The subjects were 15–18 years old (*N* = 1564, 99.6%) and were high school or secondary school students with extensive experience of smartphone adoption and usage. Moreover, 30.13% of the subjects had a history of left-behind experience.

### 3.4. Analytic Approach

Several data analyses were conducted to examine the relationships between trait mindfulness, social anxiety, and smartphone addiction. In this study, we used SPSS 26.0 for data analysis. Moreover, we used a correlation matrix to describe the current status of the adolescents’ trait mindfulness, social anxiety, and smartphone addiction. We used path analysis as part of structural equation modeling to discern the relationship between the variables. The findings from this analysis were assessed using common data-model fit statistics and their cut-off points (standardized root mean square residual and root mean square error of approximation (RMSEA) <= 0.08 and <=0.05 indicate adequate and close data-model fits, respectively; we also used the goodness of fit index (GFI) and comparative fit index (CFI)). Finally, the Model 4 procedure in SPSS PROCESS 3.5 prepared by Hayes [57] was used for the testing of mediating effects and the Model 14 procedure for moderated mediation testing, respectively. The significance level for this study was *p* = 0.1.

## 4. Results

### 4.1. Descriptive Analysis of Study Variables and Correlation Matrix

In this study, we used correlation analysis to initially explore the relationship between the variables. The descriptive analysis of each study variable and the correlation matrix are shown in Table 1. Among them, trait mindfulness was negatively correlated with social anxiety; there was a positive relationship between social anxiety and all four dimensions of smartphone addiction; trait mindfulness was also negatively correlated with all four levels of smartphone addiction (all *p*-values < 0.01).

Based on the correlation analysis, we used path analysis as part of structural equation modeling to test the relationship between the variables, and the model fit well (x^2^ = 17.696, x^2^/df = 2.95, RMSEA = 0.023, CFI = 0.981). The results of path analysis showed that trait mindfulness significantly and negatively affected social anxiety (*p* < 0.05), with a standardized effect of −0.421. Analysis of subsequent paths, in turn, revealed that trait mindfulness could significantly and negatively affect all four dimensions of smartphone addiction (all *p*-values < 0.05), with standardized effects of −0.263, −0.345, −0.201, and −0.273. At the same time, social anxiety positively affected smartphone addiction (all *p*-values < 0.05), with effect coefficients of 0.306, 0.198, 0.435, and 0.358, and the results obtained in the summary are shown in Table 2.

### 4.2. Test for Mediating Effects with Moderation

In this study, the Model 4 procedure in the SPSS PROCESS 3.5 plug-in prepared by Hayes [57] was used to conduct the test for mediating effects of social anxiety, with trait mindfulness set as the independent variable, social anxiety set as the mediating variable, and smartphone addiction set as the dependent variable. After controlling for demographic variables, such as age and gender, we determined whether the mediating effect was significant based on whether the 95% confidence interval contained 0. The results of the test are shown in Table 3. The mediating effects of social anxiety on each dimension of trait mindfulness and smartphone addiction were significant, with mediating effect values of −0.118, −0.197, −0.067, and −0.156 (none of the confidence intervals contained 0), accounting for 29.06%, 44.67%, 17.27%, and 34.82% of the direct effects, respectively. After controlling for the mediating variable social anxiety, the effect of trait mindfulness on adolescent smartphone addiction remained significant (all *p*-values < 0.05), with effect values of −0.406, −0.441, −0.388, and −0.448, respectively; therefore, social anxiety plays a partially mediating role.

### 4.3. Test for Mediating Effects with Moderation

The moderated mediating effects test was then completed using the Model 14 procedure in the SPSS macro prepared by Hayes [57]. Model 14 assumes that the second half of the mediating effect is moderated, which was more consistent with the hypothesized model in this study. After controlling for demographic variables, such as gender and age, the moderated mediating model was constructed with social anxiety as the mediating variable and left-behind experience as the moderating variable, and the test results are shown in Table 4. After adding left-behind experience to the model, the product of the interaction between social anxiety and left-behind experience had a significant predictive effect on the propensity toward smartphone addiction (β = 0.092, *p* < 0.05), indicating that left-behind experience had a significant role and trait mindfulness plays a moderating role in influencing smartphone addiction.

To understand the essence of the moderating role, we conducted a simple slope analysis, and the results are shown in Figure 2. For adolescents with left-behind experience, social anxiety had a more significant predictive effect on smartphones (β = 0.482, *p* < 0.05). For adolescents without left-behind experience, social anxiety also affected smartphone addiction, but its predictive effect was relatively small (β = 0.385, *p* < 0.05). This suggests that this moderating effect is more significant for adolescents with left-behind experience than those without it. This also directly confirms that left-behind experience is a risk factor for adolescents’ internalizing problems.

The moderated mediating effect was further analyzed. For adolescents with left-behind experience, the mediating effect of social anxiety on the positive thinking trait and smartphone addiction was −0.173 (95% CI (−0.210–0.135)), accounting for 39.23% of the direct effect, while for adolescents without left-behind experience, the mediating effect of social anxiety was −0.128 (95% CI (−0.157–0.101)), accounting for 29.02% of the direct impact. Thus, the partial mediating effect for adolescents with left-behind experience was more significant compared to that for adolescents without left-behind experience.

## 5. Discussion

### 5.1. Discussion of Findings

#### 5.1.1. The Inhibitory Effect of Trait Mindfulness on Smartphone Addiction

The empirical results of the present study showed that trait mindfulness negatively influenced the dimensions of smartphone addiction, suggesting that trait mindfulness is a protective factor against smartphone addiction in adolescents and can effectively inhibit the tendency to become addicted to smartphones, supporting research hypothesis 1. This is consistent with previous research findings and supports the hypothesis that mindfulness is a positive mental resource for individuals, representing the individual’s ability to remain aware and focused on present-moment experiences [58]. It can be posited that adolescents with elevated levels of trait mindfulness are more prone to experiencing the present moment with heightened focus and attentiveness towards internal perception. This cognitive style helps promote the positive construction of self-perception and diminishes the tendency for addictive smartphone behavior. At the same time, individuals with high levels of trait mindfulness have a greater sense of “letting go” of negative experiences and behaviors when confronted with them [59]. They can focus on the experience more healthfully and productively, thereby improving the negative behavior related to smartphone addiction. Thus, increasing the level of trait mindfulness can weaken the tendency for individuals to become addicted to smartphones.

#### 5.1.2. The Mediating Role of Social Anxiety

The present study further revealed the partial mediating role of social anxiety in the association between trait mindfulness and smartphone addiction, thereby affirming research hypothesis 2. On the one hand, congruent with the compensatory Internet use theory predictions [60], adolescents suffering from social anxiety tend to favor online socialization over in-person interactions to fulfill their interpersonal needs. However, compensatory media use motives are often counterproductive, and excessive reliance on online socialization may lead to the development of smartphone addiction. Empirical findings demonstrate a noteworthy positive correlation between social anxiety and diverse dimensions of smartphone addiction. Social anxiety, as a typical manifestation of impaired psychosocial development, stands as a crucial risk factor in triggering smartphone addiction [61]. Fear of judgment by others, self-focused attention during interpersonal interactions, and the existence of cognitive processing biases are among the prominent attributes of social anxiety, as they increase apprehension while interacting with others [62]. Smartphones exhibit the characteristics of anonymity and strong interactivity in mediated interactions, thereby, to some extent, weakening the possibility of socially anxious individuals’ cognitive bias being influenced by the negative expressions of others. Simultaneously, smartphones allow socially anxious individuals to conceal or regulate their negative emotional symptoms, increasing their comfort level in interactions. Hence, for socially anxious adolescents, online communication may transform into a replacement for interpersonal communication [47], thereby somewhat intensifying their dependence on smartphones.

On the other hand, high levels of trait mindfulness contribute to weakening social anxiety; i.e., trait mindfulness negatively affects social anxiety. Mindfulness is a cognitive model of “re-perception” [16]. This particular mode of cognition helps individuals be aware of undesirable self-views that operate automatically at the unconscious level and look at the self and accept it more objectively from a spectator’s perspective. This can precisely counteract social anxiety [63] and prevent further processing or anxiety related to the unpleasant experience. At the same time, mindfulness also contains attitudinal factors, such as openness, curiosity, and acceptance, which enable individuals to positively mediate undesirable factors such as social anxiety [64] and de-automate their emotions. Therefore, trait mindfulness, as an effective self-regulation ability for individuals, can protect individuals from internalizing problems by weakening their social anxiety. This finding indicates that, when addressing smartphone addiction in adolescents, we can start with their social anxiety and focus on cultivating trait mindfulness.

#### 5.1.3. The Moderating Effect of the Left-Behind Experience

The study found that left-behind experience moderated the second half of the mediating chain of “trait mindfulness-social anxiety-smartphone addiction”. Specifically, the correlation between social anxiety and smartphone addiction was more pronounced in adolescents with left-behind experience, which confirmed that left-behind experience is a moderator of smartphone addiction in adolescents, and research hypothesis 3 was tested.

Qualitative stress theory suggests that individuals with certain risk or vulnerability genes are more likely to be affected by adverse environmental factors and develop psychological or behavioral problems. Thus, the primary stressors for adolescents with left-behind experience stem from early adverse experiences, such as indifferent parenting, abuse, and peer aggression [65]. In coping with these risks and stresses, such adolescents consume positive psychological resources (e.g., trait mindfulness). Moreover, they have fewer positive psychological resources due to the general lack of protective factors, such as effective parental supervision, proper guidance, and social attention [66]. In addition, in an environment lacking protective factors, the combination of left-behind adolescents’ personality factors and excessive anxiety is highly likely to have long-lasting adverse effects on their positive emotions, self-efficacy, and environmental adaptability [67]. As they enter secondary school and gradually become exposed to and integrated into society, it is difficult for their weaker resilience and self-regulation to protect them against psychosocial problems in the face of a drastically changing environment, which, in turn, is more likely to result in externalizing problems, such as smartphone addiction.

Resource conservation theory suggests that individuals can mobilize positive psychological resources to cope with the adverse effects of risk and stress factors. For example, trait mindfulness, as a positive psychological resource, has a mitigating effect on perceived stress and negative emotions in daily life, and treating the left-behind experience in childhood as an event in the mind, without a negative evaluation, protects individuals’ psychological health. However, the positive psychological resources of adolescents with left-behind experience are more limited. Resource inputs and acquisitions usually fall short of what is needed, making it easy to fall into a spiral of resource loss [68]. This similarly diminishes the buffering effect of trait mindfulness on social anxiety, affecting smartphone addictive behavior.

### 5.2. Theoretical Contributions

This study offers two main theoretical contributions. Firstly, prior research has focused primarily on smartphone addiction in general adolescent populations, while marginalized adolescent groups have been less frequently included in studies. Due to the lack of parental presence and emotional companionship [69], adolescents who have experienced prolonged left-behind situations are more susceptible to smartphone addiction. Past studies have explored the addictive behaviors of left-behind children, but these studies primarily focus on their everyday experiences and lack a certain developmental perspective. Research suggests that left-behind experience, as a risk factor in early adulthood, may differ somewhat from that in a risky environment for left-behind children regarding the relationship between social anxiety and smartphone addiction. In contrast, this study incorporates a personal developmental perspective to examine how left-behind experience subliminally influences addictive behaviors in early adulthood, and the findings have implications for understanding the mechanisms underlying smartphone addiction.

Secondly, this study enriches the literature on the individual characteristics of smartphone addiction behavior by investigating how trait mindfulness influences smartphone addiction in adolescents who have experienced prolonged left-behind situations. This study offers insights into the compensatory Internet use and conservation of resources theories at the individual-trait level. On the one hand, consistent with the basic connotations of the conservation of resources theory, trait mindfulness as a positive psychological resource can counteract social anxiety situations and thus alleviate smartphone addiction behaviors. On the other hand, left-behind experience places these adolescents in a relatively disadvantaged position, making them more likely to fall into resource-deprivation situations. The findings of this study also extend the “resource loss spiral” of the conservation of resources theory [68].

Specifically, this study found that social anxiety has a stronger predictive effect on smartphone addiction among adolescents who have experienced prolonged left-behind situations. Left-behind experience is one of the factors that hinder psychosocial development, and adolescents with left-behind experience have difficulty repairing social anxiety, often turning to online socialization to compensate for their lack of offline socialization. This can lead to developmental disruptions and subsequent pathological compensation processes if the situation is not improved [70]. This finding helps explain why adolescents motivated by compensatory Internet use are more likely to develop addictive behaviors [71]. As such, this study validates the connotations and principles of the conservation of resources theory and enriches and expands the applicable contexts of compensatory Internet use theory.

### 5.3. Practice Inspiration

In terms of practical implications, this study offers a practical approach for mitigating adolescent smartphone addiction through empirical analysis. By identifying the risk factors and boundary conditions of smartphone addiction, it may be possible to reduce the incidence of such addictive behaviors among adolescents. The first dimension relates to the risk factor of social anxiety in adolescent smartphone addiction, as revealed by this study. Consistent with the findings, interventions to mitigate smartphone addiction among adolescents should involve increasing their trait mindfulness. Fortunately, unlike other trait variables, trait mindfulness is malleable and can be enhanced through cognitive therapies, such as mindfulness training [72]. Based on mindfulness awareness and attitudes, individuals can learn to perceive negative reactions and anxiety as temporary and subsequently develop strategies to cope with such undesirable emotions, a process known as benign risk [73]. Changing adolescents’ poor emotion regulation habits can also alleviate smartphone addiction. For instance, directing adolescents to focus on daily activities to replace smartphone use behaviors can effectively mitigate their digital technology use [74]. When individuals’ social anxiety levels are diminished, they tend to be more proactive in interpersonal interactions and gain social satisfaction, thereby reducing their smartphone addictive behaviors. Thus, the higher the level of adolescent mindfulness training, the more effective it is in mitigating social anxiety, thereby interrupting the “social anxiety-smartphone addiction” chain and providing a possible buffer against smartphone addiction.

More importantly, this study suggests that left-behind experience is a boundary condition for adolescent smartphone addiction. It is argued that the lack of emotional family companionship during adolescence leads to the development of withdrawn, closed, and vulnerable personalities that perceive interpersonal relationships as exclusive or insecure, making it more difficult to establish peer relationships, friendships, and intimate relationships throughout the lifespan, resulting in lasting social deficits and maladjustment. The left-behind experience during childhood leaves adolescents lacking positive psychological resources and, in a relatively vulnerable environment, they are more likely to adopt negative coping strategies or seek online compensation, thus falling into smartphone addictive behaviors. This suggests that, when implementing interventions for smartphone addiction, it is crucial to focus on adolescents with left-behind experience, create a supportive social environment, and guide individuals to reduce attrition, enhance resources through stress release and psychological guidance, and cultivate qualities such as optimism and hope [75] so that individuals can better cope with internal and external risks.

### 5.4. Research Gaps and Prospects

The present study has several limitations that need to be acknowledged. Firstly, the MAAS scale used to measure trait mindfulness in this study is a one-dimensional scale that cannot express the multidimensional conceptual structure of mindfulness. We could not explore the effects of social anxiety and smartphone addiction more deeply. Future studies may consider diversifying the dimensions of mindfulness. Second, although this study met our expectations for a specific relationship, as a cross-sectional study, it only discussed the effects of trait mindfulness and social anxiety on smartphone addiction. There may be interactions between these variables, and the cultivation of trait mindfulness may weaken individuals’ social anxiety levels and thus inhibit smartphone addiction. This causal link could not be proven. These relationships need to be deepened through detailed qualitative studies and longitudinal experiments. Future studies may consider conducting longitudinal experiments for data collection to enrich and support the results of this study.

More importantly, the data in this study were obtained through self-report measures, and the sample was limited to one province, which may have affected the generalizability of the results. Future studies may consider using a broader sample and integrating data from multiple sources, such as peers and parents, to improve the reliability and validity of the results. In addition, left-behind experience, as a risk factor in early adolescence, may differ from the experience of left-behind children who are currently in a risky environment in terms of the relationship between social anxiety and smartphone addiction. Therefore, future studies can employ a longitudinal research design examining the relation between left-behind children–left-behind experience in adolescence to further verify whether there is an inversion of the immediate and delayed effects of left-behind experience as a risk factor.

## 6. Conclusions

In this study, we elucidate how trait mindfulness can function as a buffer against social anxiety, thereby weakening the tendency towards smartphone addiction. Furthermore, we explored the contextual factors that moderate the “social anxiety-phone addiction” pathway; specifically, the role of left-behind experience. By shedding light on the complex interplay between trait mindfulness, social anxiety, and smartphone addiction, we offer valuable empirical evidence that can aid in accurately identifying the risk factors and contextual factors associated with adolescent smartphone addiction. Ultimately, our findings can inform early identification and scientific intervention strategies for adolescents who may be at risk of experiencing internalization problems.

## Figures and Tables

**Figure 1 behavsci-13-00540-f001:**
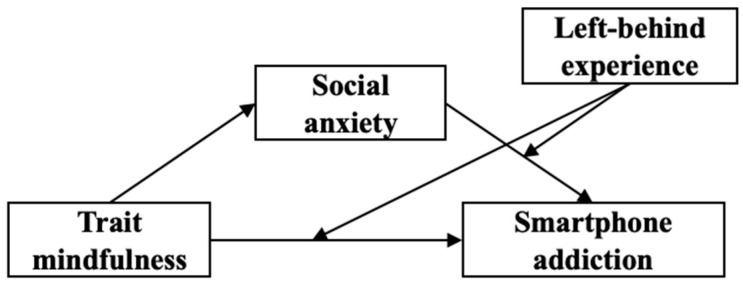
A model of the predictors of adolescent smartphone addiction.

**Figure 2 behavsci-13-00540-f002:**
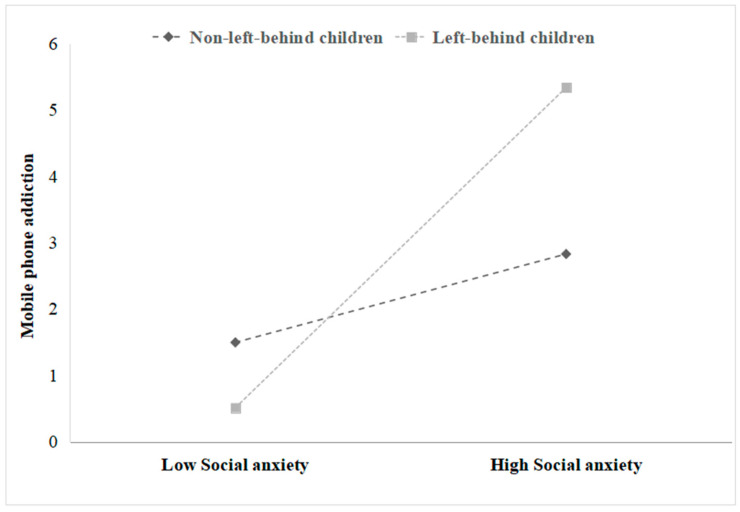
The moderating effect of left-behind experience.

**Table 1 behavsci-13-00540-t001:** Means, standard deviations, and correlation coefficients for the variables (*n* = 1570).

	M	SD	1	2	3	4	5	6
1. Trait mindfulness	3.361	0.955	—					
2. Social anxiety	3.303	0.972	−0.421 **	—				
3. Withdrawal symptoms	2.781	0.945	−0.392 **	0.416 **	—			
4. Salient behaviors	2.226	0.823	−0.429 **	0.343 **	0.802 **	—		
5. Social soothing	2.829	1.083	−0.384 **	0.520 **	0.628 **	0.547 **	—	
6. Mood changes	2.660	0.996	−0.424 **	0.473 **	0.725 **	0.686 **	0.585 **	—

Note: ** *p* < 0.01.

**Table 2 behavsci-13-00540-t002:** Path coefficients between study variables.

X	→	Y	Standardized Effect	SE	*p*	Hypothesis Validation
Trait mindfulness		Social anxiety	−0.421	0.023	0.007	Support
Social anxiety		Withdrawal symptoms	0.306	0.024	0.002	Support
Social anxiety		Salient behaviors	0.198	0.021	0.021	Support
Social anxiety		Social soothing	0.435	0.026	0.048	Support
Social anxiety		Mood changes	0.358	0.024	0.014	Support
Trait mindfulness		Withdrawal symptoms	−0.263	0.024	0.018	Support
Trait mindfulness		Salient behaviors	−0.345	0.021	0.036	Support
Trait mindfulness		Social soothing	−0.201	0.026	0.012	Support
Trait mindfulness		Mood changes	−0.273	0.024	0.014	Support

**Table 3 behavsci-13-00540-t003:** Testing the mediating effect of social anxiety.

Paths	Total Effect	Indirect Effect	SE	BootLLCI	BootULCI
Trait mindfulness–social anxiety –withdrawal symptoms	−0.406 **	−0.118	0.009	0.093	0.153
Trait mindfulness–social anxiety –salient behaviors	−0.441 **	−0.197	0.011	0.129	0.195
Trait mindfulness–social anxiety –social soothing	−0.388 **	−0.067	0.008	0.078	0.127
Trait mindfulness–social anxiety –mood changes	−0.448 **	−0.156	0.011	0.107	0.163

Note: CI = confidence interval; SE = standard error; ** *p* < 0.01.

**Table 4 behavsci-13-00540-t004:** Test for mediating effects with moderation.

	b	SE	t	*p*	LLCI	ULCI
Constants	2.729	0.130	21.070	0.000	2.475	2.983
Trait mindfulness	−0.312	0.023	−13.780	0.000	−0.356	−0.267
Social anxiety	0.267	0.023	−2.310	0.000	0.222	0.312
LBE	−0.304	0.132	−2.306	0.021	−0.563	−0.046
LBE * social anxiety	0.092	0.038	2.427	0.015	0.018	0.167

Note: LBE = left-behind experience.

## Data Availability

The data presented in this study are available on request from the corresponding author.

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
