# Peer review of "How Does Trait Mindfulness Weaken the Effects of Risk Factors for Adolescent Smartphone Addiction? A Moderated Mediation Model"

_behavsci, 2023, doi:10.3390/bs13070540_

Round 1
Reviewer 1 Report
The present paper examines the relationship between trait mindfulness and smartphone addiction in a sample of adolescents in China while exploring the role of social anxiety as a mediator in this relationship. Furthermore, the experience of being left/behind was also examined as moderator of the link between social anxiety and smartphone addiction. While the authors marshal a great bulk of research data in support of their model, I feel that the proposed mediation moderation model lacks conceptual clarity.
For one, it is not clear why should social anxiety mediate effects of trait mindfulness on smartphone addiction? I would rather hypothesize the contrary, that trait mindfulness could be considered as a mediator/moderator of the positive relationship between social anxiety and smartphone addiction. This is in line with the literature well exposed by authors in their paper showing that social anxiety depletes cognitive resources hence can lead to an increased addictive behavior, while trait mindfulness might serve as a buffer in this relationship given its strong link to self-regulatory behavior. Even the title chosen by the authors would suggest this conceptual model rather than the one they propose How does trait mindfulness weaken the effects of risk factors for adolescent smartphone addiction?
Secondly, it is not clear why experiences of being left behind should moderate rather than mediate the effects of social anxiety on smartphone addiction?
I suggest authors revise their conceptual model and examine the role of trait mindfulness as being left behind as a mediators of the relationship between social anxiety and smartphone addiction (see for instance https://doi.org/10.1007/s10459-021-10039-w).
Minor comments
Authors should provide full references for instruments used in the study. Es. publication year is missing in almost all references provided for the measures.
A conclusion section is missing.
Author Response
Dear reviewers,
We sincerely appreciate your professional feedback on our manuscript. We acknowledge the need for further conceptual definition of the study variables and a more comprehensive explanation of the relationships between these variables to enhance the quality of the paper.
1.Regarding your concerns about why social anxiety mediates the effect of trait mindfulness on smartphone addiction and why the left-behind experience plays a moderating role, we have taken your suggestions into careful consideration. In order to provide a clearer response, we have developed a simple model diagram that illustrates the subsequent derivation process. Please see the attachment for details.
2. We would like to express our sincere apologies for the oversight in not providing the reference to the study instrument as you rightfully pointed out. We understand that the source of the scale used in our study is crucial, and we deeply regret this omission. To rectify this, we have included the references to the research instrument in our manuscript. Specifically, you can find the basis of the research instrument in Articles 54, 55, and 56, as indicated in the reference section.
3.Furthermore, we have carefully reviewed the journal's manuscript format guidelines and made the necessary adjustments. As per your suggestion, we have brought forward the Discussion section to ensure its proper placement within the article. Additionally, we have added a concise conclusion at the end of the manuscript to enhance the overall integrity and coherence of the article structure.
We would like to express our sincere gratitude for your valuable suggestions, as they have played a crucial role in improving the quality of our manuscript. We have taken great care to address and incorporate your requests in a detailed manner. We kindly ask you to review the revised manuscript again and provide further feedback.

Reviewer 2 Report
The author of this paper made a very reasonable study to explain the relationship between crucial variables in left-behind adolescents. The findings and proposal of models are quite interesting. These are my suggestions:
1. There are several citations in the text that do not appear in the reference list. Please provide the complete references at the end.
Examples: Deng, et al (page 6 second paragraph), Xiong, et al page 6, third paragraph; Hayes (2013), page 7 first paragraph; also page 8 first and second paragraphs.
2. Authors used two different reference citations, please use the adequate one for the journal.
3. The authors do not explain if the scales were o not validated in the local language. Please talk about it in the text. (page 6).
4. The authors reported on pages 7 and 8 that "trait mindfulness could significantly and negatively affect all four dimensions of smartphone addiction (all P values<0.05) with standardized effects of -0.263, -0.345, -0.228, and -0.285. At the same time, social anxiety positively affected smartphone addiction (all P-values<0.05), with effect coefficients of 0.297, 0.167, 0.485, and 0.367, and the results obtained in the summary are shown in Table 2".
a) Please explain o justify the POSITIVE 0.358 value of trait mindfulness versus mood changes, which was presented as a NEGATIVE EFFECT.
b) Also, explain why the values they are reporting in the text do not check with the values in Table 2.
5. Please, perform a careful revision of the references cited in the text, and in the final reference list.
Author Response
Dear reviewers,
We would like to express our sincere gratitude for your recognition of our article and your valuable professional comments. We have thoroughly reviewed the details of the paper, taking into account your suggestions, and have made the necessary corrections accordingly.
1.With regard to the missing references you mentioned, we have conducted a thorough check and have included the appropriate additions. You can now find the relevant references in the revised manuscript, specifically in references 54, 55, and 57.
2.Regarding the reference format, we have used Zotero once again to generate the references in the required format specified by the journal. We carefully reviewed and modified each reference to ensure their accuracy and adherence to the correct format in the newly submitted version.
3.For the English scale used in the study, we took significant measures to ensure its reliability and validity. We engaged the expertise of two professional English graduate students for the translation process, and a psychology professor validated the content validity of the scale. The adaptation of the scale was carried out diligently, closely aligning it with the study scenario. As for your query about the use of the local language for validation, we ensured that the Chinese language versions no any bias in understanding the scale. Additionally, we provided an introductory interpretation before administering the scale to the participants to facilitate a more accurate comprehension of its content.
4.We deeply apologize for the inconsistency in the data reporting on pages 7 and 8, which was a result of the iteration of data and the inclusion of both standardized and unstandardized coefficients in each version. We acknowledge that we should have been more attentive during the final check, which led to the incorrect transcription of the data. In the latest uploaded version, we have meticulously reviewed the data to ensure uniformity and accuracy, as described in the first paragraph on page 7-8.
We sincerely appreciate your invaluable suggestions, as they have played a vital role in improving the quality of our paper. We have implemented the necessary revisions and adjustments based on your feedback.
Round 2
Reviewer 1 Report
Authors have addressed all comments raised in my review.